

Precipitable Water Characteristics during the 2013 Colorado Flood using Ground-Based
GPS Measurements
Hannah K. Huelsing[1,2], Junhong Wang[2], Carl Mears[3], and John J. Braun[1]
1. Constellation Observing System for Meteorology, Ionosphere, and Climate
Program Office
University Corporation for Atmospheric Research
3300 Mitchell Lane, Boulder, CO 80301
2. Department of Atmospheric and Environmental Sciences
University at Albany, SUNY
1400 Washington Avenue, Albany, NY 12222
3. Remote Sensing Systems
444 10[th] Street, #200
Santa Rosa, CA 95401
Corresponding Author:
Hannah K. Huelsing
Constellation Observing System for Meteorology, Ionosphere, and Climate Program
University Corporation for Atmospheric Research
3300 Mitchell Lane, Boulder, CO 80301
Phone: (303) 497-2606; Email: huelsing@ucar.edu





**Abstract**

During 9th-16th September 2013, the Front Range region of Colorado experienced

heavy rainfall that resulted in severe flooding. Precipitation totals for the event exceeded
450 mm, damages to public and private properties were estimated to be over \$2 billion,
and nine lives were lost. This study analyzes the characteristics of precipitable water
(PW) surrounding the event using 10 years of high-resolution GPS PW data in Boulder,
Colorado, which was located within the region of maximum rainfall. PW in Boulder is
dominated by seasonal variability with an average summertime maximum of 36 mm. In
2013, the seasonal PW maximum extended into early September and the September
monthly mean PW exceeded the 99th percentile of climatology with a value 25% higher
than the 40 year climatology. Prior to the flood, around 18 UTC on 8 September, PW
rapidly increased from 22mm to 32mm and remained around 30mm for the entire event
as a result of the nearly saturated atmosphere. The frequency distribution of September
PW for Boulder is typically normal, but in 2013 the distribution was bimodal due to a
combination of above average PW values from September 1st-15th and much drier
conditions from 16th-30th September. The above normal, near saturation PW values
during the flood were the result of large-scale moisture transport into Colorado from the
eastern tropical Pacific and the Gulf of Mexico. This moisture transport was the product
of a stagnating, cutoff low over the southwestern United States working in conjunction
with an anticyclone located over the southeastern United States. A blocking ridge located
over the Canadian Rocky Mountains kept both of the synoptic features in place over the
course of several days, which helped to provide continuous moisture to the storm, thus
enhancing the accumulated precipitation totals.



**Keywords**
Precipitable Water, GPS, 2013 Colorado Flood, Extreme Precipitation























## 1. Introduction


During 9th-16th September 2013, multiple local and state precipitation records
were broken when low-level, easterly flow interacted with an anomalous moisture pool
over the Front Range region of Colorado to produce one of the largest floods in state
history (Colorado Climate Center, 2013). The heaviest and most persistent rainfall
occurred on the 11th and 12th of September, with a maximum centered over Boulder and
Larimer counties (Fig. 1). In the hardest hit areas, total precipitation accumulation
exceeded 450 mm (17.7 in) (Gochis et al. 2015). The city of Boulder set multiple records,
observing 292.6 mm over the course of two days and 341.8 mm over the course of three
days. The resultant flooding claimed nine lives and caused 1,100 documented landslides.
Damages to public and private properties were estimated to be over $2 billion (Gochis et
al. 2015).
The following summary of the September 2013 event was first presented in
Gochis et al. (2015). Surface temperatures were in the 16-18 °C (60-64 °F) range and
precipitable water (PW) values were high. Periods of heavy precipitation exceeding
25 mm (1 in) per hour, along with flooding, began on the evening of 11th September, with
the heaviest portions over the Front Range, the area outlined in Fig. 1. The mountainous
region between Boulder and Estes Park experienced the heaviest rain rates, which ranged
from 25-50 mm (1-2 in) per hour and resulted in an overnight total exceeding 200 mm
(8 in). Somewhat lighter rainfall continued into the 12th, becoming intense once more
during the afternoon hours and increasing rainfall totals to over 380 mm (15 in) in the
Boulder to Estes Park region. By the 13th, precipitation had finally lessened to
intermittent showers and widespread drizzle, finally clearing on the 14th. A final surge of





moisture occurred on the 15th and resulted in 25-50 mm (1-2 in) of widespread, moderate
rainfall on soils that were already saturated, thus increasing the amount of runoff.

95        This event was uncharacteristic, not only because of its rainfall amounts but also

because of the time of year in which it occurred. Petersen et al. (1999) examined the
climatology of precipitation events over the Front Range region and found that, while a
majority of events occur between April and October, the convective classification of the
events differ depending on what time of year they occur in. There are two peaks in the
event distribution, the first of which occurs in late May to early June. Precipitation events
during this time are synoptic, or large, scale and quasi-stationary. The precipitation in
these events is enhanced orographically and locally, and is typically widespread and of
moderate intensity. The second peak in precipitation events occurs from late July into
early September with a pronounced maximum frequency from late July into early
August. The storms in these events generally have a small areal extent and are highly
convective. The September 2013 event was quasi-stationary and synoptic with
precipitation controlled by localized and orographic enhancements. The areal extent of
the 2013 event was large and the rainfall was of moderate intensity. According to the
climatology completed by Petersen et al. (1999), this type of event was more typical of
storms which occur in late May to early June. However, this event occurred at a time of
year when precipitation tends to be highly convective and of small areal extent, so the
timing, as well as the amount of rainfall, was abnormal.

In another study which examined the climatology of rainfall events in Colorado,

Mahoney et al. (2015) found that the region of Colorado east of the Continental Divide
does not generally experience heavy precipitation events in the fall because it is during




this time of year that the region experiences seasonal atmospheric drying. They did note
that there was enhanced climatological variability in September and October, making it
difficult to place these months into the same category as the drier months (November-
February). In general, east of the Continental Divide experiences most of its precipitation
in the spring and summer months, with the Front Range receiving a majority of its
moisture in the spring. However, extreme precipitation events are not limited to these
seasons and can also occur in fall and winter months.

Flooding due to extreme precipitation events can occur at any time of the year

because all elevations in all seasons are prone to experiencing heavy precipitation. This is
partially represented by the dates in Table 1, which compares the September 2013 event
to previous heavy precipitation events in Northern Colorado history that resulted in
catastrophic flooding (Colorado Climate Center; Maddox et al. 1977; Petersen et al.
1999; Gochis et al. 2015). Prior to the September 2013 event, there were 5 events on
record that were classified as comparable to the 2013 event by the Colorado Climate
Center. However, all except one of these storms took place in the spring and summer
months, as would be expected from the climatology of the rainfall events presented in
earlier.

Out of the events listed in Table 1, the Colorado Climate Center noted that the

event that occurred on 1st-12th September 1938 near Fort Collins, Colorado was the most
similar in timing and magnitude to the September 2013 event. Observers recorded 8-10
inches of rainfall and the surrounding region experienced severe flooding. However,
there is not much else known about this event because the amount of recorded
atmospheric data available from this time period is limited. Comparing the September





2013 event to the 5 previous events in Table 1, this event had the highest total rainfall and
caused the most damage, as is seen by the total cost of the event. This event also had a
vast areal coverage, with heavy precipitation occurring from Denver all the way into
southern Wyoming. Flooding took place as far to the east as Nebraska and caused a lot of
damage to infrastructure along the Front Range of Colorado.

The amount of precipitation that fell during the September 2013 event required a

large amount of moisture at a time when atmospheric moisture was beginning to decrease
from higher summer values (Mahoney et al., 2015). Moisture transport and quantity are
important aspects to evaluate when investigating heavy precipitation events. However,
there has not been a vast amount of research examining the characteristics of PW during
heavy precipitation events. Such characteristics are important to understand because they
could influence future weather and climate trends. Kunkel et al. (2013) found an
increasing trend in atmospheric PW quantities associated with extreme precipitation
events and suggested this trend could lead to an increase in storm intensity. While
Hoerling et al. (2014) noted that the September 2013 event was probably not connected
to climate change, they did find that heavy precipitation events are becoming more
frequent and Karl and Trenberth (2003) found evidence that the number of heavy
precipitation events is expected to increase with increasing global temperatures, such as
we are experiencing now. The observed and projected increase in the number of heavy
precipitation events, combined with the uncertainty of how PW contributes to
characteristics of these events, motivated an investigation of PW characteristics
surrounding the 2013 event so as to better understand the contributions of PW to an





extreme precipitation event with the objective to someday apply these results to future
research incorporating a wider variety of events.

As the aim of this research was to examine the characteristics of atmospheric PW

during the 2013 Colorado Flood, data with a high spatial and temporal resolution was
needed to resolve features within the event. GPS receivers are much more densely spaced
with a total of 236 stations over North America than the radiosonde network, which has a
total of 92 stations. The higher density of observations in the GPS network results in a
higher spatial resolution with which to analyze storm features and water vapor transport.
GPS data also has a much higher temporal resolution of anywhere from 30 minutes to
two hours, as compared to the standard, twice-daily launching of radiosondes.

The primary goal of this research was to investigate the magnitude and characteristics

of PW over the Front Range region associated with the September 2013 event. The goal
of this study was to answer the following scientific questions.

(1) What were the characteristics of PW surrounding this event? This portion of

research was focused on the examination of the temporal variability of PW, as

well as a comparison with climatology, before, during, and after the event.

(2) Where did the moisture for the 2013 event originate? To answer this question,

synoptic-scale dynamics and pre-existing conditions that led to large-scale,

continuous moisture transport were evaluated.

**2. Data and Methodology**
*2.1 Precipitable Water Datasets*

Two datasets were used to analyze PW characteristics surrounding the 2013 event.

The first of these was a two-hourly, long-term (1995-2015) PW dataset (Wang et al.





2007; Mears et al. 2015; Mears et al. 2016). The PW in this dataset is derived using 5-
minute International Global Navigation Satellite System (GNSS) Service (IGS) Zenith
Total Delay (ZTD) data. The analysis technique for the interpolation and conversion of
ZTD to PW is summarized in Wang et al. (2007) and two key variables used in the
conversion are water-vapor-weighted atmospheric temperature (Tm) and surface pressure
(Ps). ZTD is represented as the sum of the Zenith Hydrostatic Delay (ZHD), which is a
function of Ps, and the Zenith Wet Delay (ZWD), which is a function of PW and Tm. The
2-hourly PW data from Boulder became available starting in 2004.

The second PW dataset used in this study was the 30-minute SuomiNet dataset from

the Constellation Observing System for Meteorology, Ionosphere, and Climate
(COSMIC) group (Ware et al. 2000). The SuomiNet network currently consists of over
200 sites located around North America and the data are processed in near-real time from
raw GPS data, the values of which do not differ greatly from post-processed GPS data.
For this research, the standardized anomalies of the SuomiNet data were calculated by
subtracting PW at each time step from the mean and dividing this by the standard
deviation (Grumm and Hart 2001). The standardized anomaly data were gridded and
interpolated using a general kriging method to a grid box of 0.5°X0.5°. Kriging is defined
as optimized interpolation that is weighted by spatial covariance values and based on
regression against observed values of surrounding data points (Bohling 2005). This
method was chosen because of its simplicity and superior performance when compared
with the inverse distance weighting (IDW) method (Zimmerman et al. 1999; Yasrebi et
al. 2009).





*2.2 Formulation of a GPS PW Climatological Dataset*
PW data for Boulder, Colorado were chosen to evaluate the PW variability of this
region over the course of 10 years and compare this variability with that of 2013 to
improve the understanding of how the September 2013 event differed from climatology.
This region encompasses six SuomiNet stations and two IGS stations (Fig. 1a). To
examine the anomalous nature of the flood, a dataset with a length of at least 10 years of
observations was needed as a climatological standard for the analyzed region. While 10
years is not long enough for a standard climatology of 30 years as defined by the World
Meteorological Organization (WMO), GPS PW data for Boulder has only been available
since 2004. The PW time series of each GPS station was initially examined to determine
which, if any, station had a long enough data record to serve as the climatological
standard, and also to check for data outliers and data continuity. No stations were found
to have more than seven years of data and datasets that contained discontinuities were
discarded. A major issue that appeared during this analysis was that only one SuomiNet
and one IGS station had data observations during the September 2013 event, and neither
of them had a lengthy dataset. A decision was made to combine the data from different
stations in the region and make a 10-year dataset that included observations from the
flood.
The GPS PW data used to create the 10-year dataset were first quality-controlled by
using several methods defined in Wang et al. (2007). The first method used was the range
test in which the lower and upper limits of PW values were set as 0mm and 150mm,
respectively. The second quality-control method used involved using the mean and
standard deviation for each month to detect any outliers. This method required that at



least one-quarter of the data be present in order to have an adequate amount of
observations so that the statistical aspects could be deemed accurate. Individual PW
values within each month were analyzed and any values that were more than 4 standard
deviations away from the monthly mean were discarded (Wang et al. 2007). The quality
control removed 0.1% of the total data points for the station SA00 and less than 0.1% of
the total data points for the rest of the stations.

The next step in the creation of the 10-year dataset was to compare PW data among

the stations. PW is strongly dependent on elevation so any station that had an elevation
above 1,800 m was eliminated because these receivers were located too far above the
elevation of Boulder (1655 m). To remain consistent, the remaining stations were
compared to the station with the longest dataset and elevation closest to that of Boulder
(DSRC). Five stations were chosen for the merged 10-year PW dataset (Fig. 2) because
their averaged PW differences were not statistically significant from one another and the
elevation differences between all stations were less than 50 m. A more thorough analysis
of the complete dataset and its comparison with 2013 is described in Sect. 3. The
SuomiNet station, P041, also passed the statistical significance test, but did not have a
complete record of data for 2013 so could not be included in the 10-year dataset. Instead,
the 2013 PW data from P041 was used to analyze small-scale variability leading up to,
and during, the flood period because it has a higher temporal resolution (30 minutes) than
NIST (two-hourly), which was chosen for the 10-year dataset.
*2.3 Additional Datasets*

The data used as a long-term PW climatology dataset were twice-daily radiosonde

data from the Stapleton airport in Denver, Colorado extracted from the homogenized



radiosonde dataset created by Dai et al. (2011) (Fig. 1b). This PW dataset was created by
integrating specific humidity from the surface to 100hPa, is available from 1979 to 2013,
and was homogenized using an advanced statistical approach that is more thoroughly
described in Dai et al. (2011).
The primary dataset used to evaluate moisture transport was the North American
Regional Reanalysis (NARR) dataset, which is available from 1979 to the present
(Mesinger et al. 2006). The domain for NARR is North America and the horizontal
resolution is 32 km with 45 vertical layers. The NARR variables chosen for the
evaluation of moisture transport surrounding the event were the 500 hPa geopotential
height and the vertically integrated moisture flux.
**3. Precipitable Water Characteristics**
Gochis et al. (2015) noted that the atmosphere over Northern Colorado was
abnormally moist from 9th-16th September. Radiosondes captured PW values above
30 mm, an abnormal value for a semi-arid climate. Gochis et al. (2015) also noted that
the raindrop distribution during the event consisted of numerous small raindrops, which
is more commonly observed in a tropical climate. To better understand how abnormal the
atmospheric moisture was during this event, the magnitude, distributions, and variability
of PW over Boulder were evaluated and compared to climatology.
*3.1 Temporal Variability of Precipitable Water*
First, the temporal characteristics of September of 2013 were compared with the
10-year GPS PW dataset described in Sect. 2.2. Figure 2 shows the time series of the
merged 10-year PW dataset discussed in Sect. 2. The strongest PW variation is seasonal
with a mean seasonality of 18mm and the summer peaks are coincident with the annual



occurrence of the wet season in Colorado. Also note that the belted appearance of this
time series represents synoptic and diurnal PW variability, the latter of which has an
average magnitude of 8 mm. The maximum value of PW for 2013 was 33.5 mm on
September 12. Note the extension of high PW values from the summer months into early
September of 2013. This extension is not observed in any of the other years contained in
this dataset and is an indication that the atmosphere was anomalously moist for the time
of year in which the flood occurred.
Figure 3 zooms in on the extension of high PW values observed in September of 2013,
giving a clearer view of the temporal variability of PW surrounding the flood event. The
high PW values from $28^{th}$ August - $5^{th}$ September represent moisture associated with the
end of the North American Monsoon. These high values begin to decrease around $6^{th}$
September before quickly rising on September $9^{th}$ into the $10^{th}$, with values spiking to
above 30 mm. PW decreases slightly to 26 mm until the $11^{th}$, when it once again
increases to above 30 mm where it remains until the $13^{th}$. After this, PW decreases to
values closer to the September climatological average of 15 mm. An interesting point to
take note of is that PW values stay relatively constant during the event despite the fact
that continuous, and sometimes heavy, precipitation is occurring. For PW to remain at
high values over multiple days, as was seen here, moisture needed to be continuously
transported into the region (Gimeno et al. 2012). Had there not been a constant transport
of moisture, PW would have decreased as atmospheric moisture condensed and formed
precipitation. The examination of the moisture transport that fueled this event is
presented in Sect. 4.


*3.2 Precipitable Water Abnormality During the 2013 Flood*

The consistently high values of PW during the time of heaviest precipitation in

Fig. 3 led to an investigation to discern if the atmosphere over Boulder was fully
saturated during the September 2013 event. To evaluate this, observed radiosonde PW
data were compared with PW values that were calculated assuming a fully saturated
atmosphere, i.e. 100% relative humidity from the surface up to 300 hPa. Figure 4 shows
the comparison between these two variables from 6th-20th September 2013. Starting on
10th September observed and fully saturated PW values were within 5 mm of each other,
indicating an atmosphere that was very near to saturation during the course of the
September 2013 event. Except for a period of time on 14th September when the
atmosphere began to dry, observed PW stayed relatively close in value to the fully
saturated PW until 16th September.
Figure 5 compares monthly-averaged 2013 GPS data and radiosonde data to 40 and 10
years of monthly-averaged radiosonde and GPS data, respectively. 2013 PW monthly
averages were consistently lower than climatology until July. Up until July, the Front
Range was still under drought conditions according to the National Climatic Data Center
(NCDC) North American Drought Monitor. The monthly average for September of 2013
was around 20 mm, approximately 25% higher than the long-term climatological monthly
average for September. Also note that the monthly average for September of 2013 is
above the 95th and 99th percentiles, which were calculated from 40 years of monthly-
averaged radiosonde data. McKee and Doesken (1997) evaluated extreme precipitation
events for Colorado from the late 1800's up until 1996 and found that, for these events,
PW never exceeded the 95th percentile. That the monthly averaged PW for September of



2013 exceeded the 99[th] percentile when compared to 40 years of data shows just how
anomalous the event was in terms of PW magnitude and timing.

Another tool used to evaluate how anomalous the 2013 Event was in terms of PW

was to examine the PW frequency distributions. Foster et al. (2006) examined the
monthly and annual frequency distributions of PW data for various stations and found
that there were three main types of distributions for PW data: lognormal, which is the
most common distribution around the world; reverse-lognormal, which represents an
atmosphere near saturation; and bimodal, which occurs in regions with strong seasonal
variability such as monsoonal zones.

To analyze PW frequency for this event, monthly distributions were created for

June through September of 2004-2013 (Fig. 6). The skewness of each distribution was
then calculated and these values, along with visual analysis, were used to determine if
each distribution was normal, lognormal, reverse-lognormal, or bimodal. Bulmer (1979)
provided guidelines for interpreting the skewness of a distribution that were employed
when evaluating the distributions in this study. A normal distribution has a skewness
from -0.5 to 0.5, while a positive (negative) skewness with its absolute values within 0.5
to 1 represents a lognormal (reverse-lognormal) distribution (Bulmer 1979; Foster et al.

2006).

Upon analyzing the distributions in Fig. 6, June through September primarily

have normal distributions with September being, on average, slightly more positively
skewed than the other months with a value of 0.32, although the distribution is still
considered normal according to the conditions for skewness defined in Bulmer (1979).
However, the seasonal variation in PW is still evident as July and August distributions



tend to have their highest frequencies over higher values of PW than either June or
September. Also, despite most months having a normal distribution, there are four
distributions which were labeled as lognormal because they have skewness values larger
than 0.5: July 2005, September 2008, September 2010, and June 2013.

The distribution that shows the largest shift in distribution from the other years is that

of September of 2013, which had a bimodal distribution. Figure 7 shows a more detailed
comparison of September of 2013 PW data with 10 years of GPS PW data and 40 years
of radiosonde PW data. September of 2013 PW data were split up into two categories:
"Flood", which represents 1st-15th September; and "Post-Flood", which represents 16th-
30th September. Fig. 7 shows how different September of 2013 is from climatology and
also how the atmosphere during the "Flood" differed from the "Post-Flood" atmosphere.
The atmosphere during the "Flood" was highly saturated, with a peak frequency around
25 mm and PW values as high as 35 mm. The frequency distribution during this time was
normal with a skewness value of 0.175. The "Post-Flood" atmosphere had a distinct
lognormal distribution indicated by visual analysis and also by a skewness of 0.6838. The
atmosphere at this time was considerably drier, with frequency peaking at 0.9 around
7 mm of PW.
**4. Water Vapor Transport**

The occurrence of heavy precipitation such as was observed during the September

2013 event requires sufficient moisture supply to fuel it. In Sect. 3, PW was shown to
spike rapidly prior to the flood and remain at highly anomalous values for the duration of
the event. In order to more completely understand the PW characteristics of this event, it
was important to investigate where the moisture originated and what mechanisms were





controlling the moisture transport that kept the atmosphere very near to saturation for
seven consecutive days.

The moisture source and transport for the September 2013 event was briefly

investigated in previous literature. Gochis et al. (2015) noted that the sources of moisture
for the event were the Gulf of Mexico and the eastern tropical Pacific Ocean, both of
which had 1-3 °C above normal sea surface temperature (SST) anomalies. They stated
that the moisture from these regions was transported into the Front Range by a cutoff low
over the southwestern United States working in conjunction with an anticyclone over the
southeastern United States. Both of these features were kept in place for multiple days by
a blocking ridge located over the Canadian Rockies (Gochis et al. 2015). Trenberth et al.
(2015) stated that the source of moisture for the September 2013 event was only from the
eastern tropical Pacific Ocean, while Mahoney et al. (2015) claimed the moisture for the
event came primarily from the Gulf of Mexico.

Due to the slight variation of opinion on which body of water was the source of

moisture for the event, this study further investigates moisture source and transport by
examining NARR 500 hPa geopotential height and integrated water vapor flux in
conjunction with the standardized anomaly of gridded SuomiNet PW data. Five times
surrounding the event were chosen for analysis based on their proximity to rapid
fluctuations in PW (Fig. 3). The three variables listed above are plotted in Fig. 8 at each
of the five time steps.

Figure 8a-c shows the atmospheric conditions on 6[th] September at 9 UTC, prior to

the start of the event. There was a large ridge with 500 hPa geopotential heights above
596 gpm over the western half of the United States (US) (Fig. 8a) which contributed to




higher temperatures and dried the atmosphere over Boulder as seen in Fig. 8c. At that
point, there was no direct water vapor transport from either the Gulf of Mexico or the
eastern Pacific (Fig. 8b).

Moving on to 9$^{th}$ September at 18 UTC (Fig. 8d-f), a trough started to form over

the western United States and an anticyclone shifted over the southeastern US (Fig. 8d).
Together, these began transporting water vapor towards the northeast along the eastern
flank of the trough from the eastern Pacific (Fig. 8e). This transport contributed to a belt
of PW anomalies with magnitudes of 1.5 to 2.5 standard deviations over the southwestern
and western US (Fig. 8f). The PW anomaly over Boulder at that point was between 1-1.5
standard deviations and precipitation had not yet begun (Fig. 9). Water vapor appeared to
travel to Colorado from the eastern Tropical Pacific at that time (Fig 8e).

By 11$^{th}$ September at 6 UTC (Fig. 8g-i), the low pressure over the western US

deepened and formed into a cut-off low (Fig. 8g). The low stagnated over the western US
due to the influence of the blocking ridge under which it resided. The anticyclone over
the eastern US also strengthened. Working in conjunction, the strengthening of the low
and the high increased the southerly water vapor transport and there was a corridor of
flux convergence over New Mexico and the direction of the flux over Northern Colorado
was toward the Rocky Mountains (Fig. 8h). This resulted in a corridor of PW anomalies
that stretched from the Mexican border to southern Wyoming (Fig. 8i). The magnitude of
the PW anomaly over Boulder rose to between 2.5 to 3 standard deviations as the
moisture pooled against the Rocky Mountains due to easterly water vapor transport.
Light, orographically enhanced precipitation began and Boulder experienced rain rates



around 5mm h$^{-1}$ (Fig. 9). Water vapor was being transported into Colorado from the
eastern Tropical Pacific and the Gulf of Mexico at this time (Fig 8h).

By 12$^{th}$ September at 6 UTC (Fig. 8j-l), the anticyclone began to break down but

the cutoff low deepened even further (Fig. 8j). Water vapor was still being transported
into the region from the Gulf of Mexico by the synoptic conditions with an easterly
component of the flux continuing to pool water vapor against the Rocky Mountains (Fig.
8k). However, the transport of moisture into Colorado appeared to have weakened
substantially and the eastern Tropical Pacific was no longer a source of moisture. There
was still a corridor of PW anomalies coinciding with the regions of strong water vapor
flux and the magnitude of the anomaly over Boulder was still between 2.5 to 3 standard
deviations (Fig. 8l). Precipitation intensified over the past 24 hours and Boulder
experienced up to 35mm h$^{-1}$ of rainfall (Fig. 9). While a majority of the rainfall was
orographically-enhanced, the occasional intense periods of rainfall were a result of
mesoscale circulations, as was noted by Gochis et al. (2015).

By 14$^{th}$ September at 21 UTC (Fig. 8m-o), the blocking ridge broke down, which

allowed synoptic conditions to shift eastward, and the cutoff low once again became a
trough (Fig. 8m). This resulted in the water vapor flux also shifting eastward (Fig. 8n).
The PW anomaly over Boulder decreased to between 1 to 2 standard deviations (Fig. 8o).
Rainfall for the event ended at this point, excluding a peak that occurred during the
afternoon of 15$^{th}$ September (Fig. 9).

Upon comparing NARR integrated moisture flux with 500 hPa geopotential

height and observed standardized PW anomalies, it was found that the strength and
location of moisture transport varied over the course of the event. Prior to the event, on



9th September, moisture from the eastern tropical Pacific appears to have been transported
up to Colorado by a stagnating cutoff low over the southwestern US. Starting on 10th
September, the cutoff low and subtropical anticyclone promoted southerly flow into
Colorado from the eastern tropical Pacific and the Gulf of Mexico. As of the 12th
September, the eastern tropical Pacific no longer provided moisture for the event and the
Gulf of Mexico was the sole source of moisture. By the 14th September, the transport of
moisture into Colorado had significantly weakened due to the eastward shift of the
synoptic pattern. The moisture transport was dependent on the strength and location of
the dominant synoptic features, and based on the analysis shown in Fig. 8 the moisture
has been transported into Colorado from both the Eastern Tropical Pacific and the Gulf of
Mexico. These results are most consistent with the findings of Gochis et al. (2015), but
do not discount the results found in Trenberth et al. (2015) and Mahoney et al. (2015).
**5. Conclusions**

The aim of this research was to analyze PW characteristics surrounding the

September 2013 event and compare them to climatology. Monthly averaged PW values
in the GPS dataset for September of 2013 was above the 99th percentile when compared
to the climatological data as well as around 25% higher than the monthly-averaged
climatological mean value for September. That the monthly average for September of
2013 was so far above the climatology for 10 and 40 years of data indicates how
anomalous the atmospheric moisture content was during the event. The frequency
distribution of PW for September of 2013 was bimodal, which was much different than
the typical normal distribution observed in September of other years. Upon further
analysis, it was noted that the highly saturated portion of the bimodal distribution was



solely the result of the September 2013 event, which had a nearly saturated atmosphere.
The second half of September had a lognormal distribution, representing a much drier
atmosphere for the rest of the month. The moisture for the event originated from the
eastern tropical Pacific at the beginning of the event 9th September, came from this source
and the Gulf of Mexico during the heaviest precipitation (10th – 12th September), and then
from only the Gulf of Mexico towards the end (12th-14th September).
**Code Availability**
Code is available from the lead author upon request.
**Data Availability**
Two-hourly GPS PW data is available upon request from the first and second authors.  30
minute SuomiNet GPS PW data is available for download in ASCII and NetCDF format
from the COSMIC group website (suominet.ucar.edu). The twice daily, homogenized
radiosonde data is available upon request from the second author. NARR data is available
for download on the National Oceanic and Atmospheric Administration (NOAA) website
(nomads.ncdc.noaa.gov/data/narr). The 1-hourly rain gauge data is available upon request
from the National Center for Atmospheric Research (NCAR) Research Applications
Laboratory (RAL).
**Author Contribution**
The first author was the primary researcher with constant assistance and guidance from
the second author. The third author was the PI on the grant and a contributing editor. The
fourth author served as an editor.
**Competing Interests**
There are no competing interests from any of the authors.





**Disclaimer**

There is no disclaimer regarding the research completed in this paper.

**Acknowledgements**

This research was supported by the National Aeronautics and Space Administration
(NASA) RSS Subcontract #6003 under the Prime Contract NNX11AO25A. The first
author would like to thank Heather Davis, Rebecca Steeves, Sarah Ditchek, Molly Smith,
Rich and Brenda Dixon, Michael Fischer, Joshua Alland, Matthew Vaughan, Casey
Peirano , Eric Adamchick, and Ted Letcher for their valuable input and support.

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

comparison of ordinary and universal Kriging and inverse distance weighting.

Mathematical Geology, 31 (4).









| Date | Location Most Affected | Total Rainfall (inches) | Deaths | Cost |
|---|---|---|---|---|
| September 1-12, 1938 | Fort Collins | 8-10 | N/A | N/A |
| May 4-9, 1969 | West of Denver | 6-9 | 0 | $136.5 million |
| July 31-August 1, 1976 | Estes Park | 12-14 | 144 | $348.5 million |
| July 27-August 4, 1997 | Fort Collins | 14.5 | 5 | $290 million |
| April 29-30, 1999 | Northern Colorado | 8-10 | 0 | $140 million |
| September 9-16, 2013 | Boulder | 16 | 8 | $2 billion |


**Table 1.** A comparison of the September 2013 Event to previous flood inducing,
heavy precipitation events in Northern Colorado history. All monetary values were
calibrated to 2013 values.



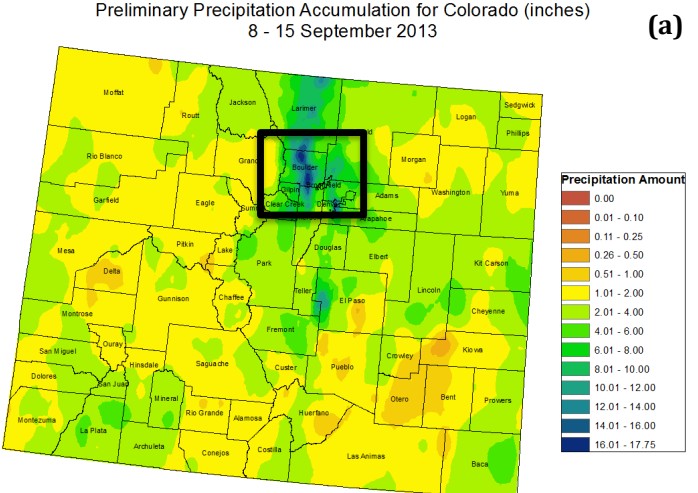


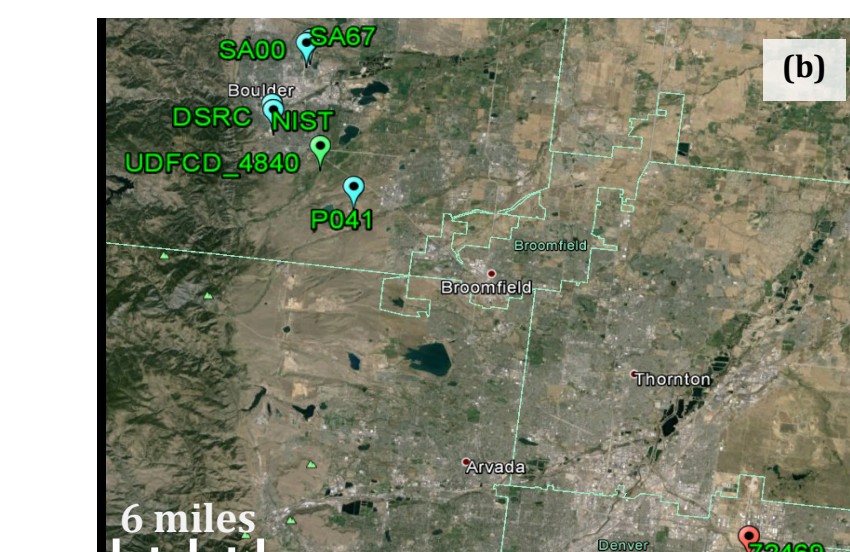

**Figure 1.** (a) Map of accumulated precipitation over Colorado from 8-15 September
2013 (image courtesy of the Colorado Climate Center) with the area depicted in (b)
outlined in the black box; and (b) the locations of the primary GPS (blue circles),
rain gauge (green circle), and radiosonde (red circle) observations used in this
study. NISU and NIST are the only IGS GPS stations plotted on this map. All of the
other GPS stations are from the SuomiNet network.






**Figure 2.** A time series of the GPS PW data for Boulder, Colorado from 2004-2013
with each station denoted by a different color, the monthly means denoted by the
solid, black line, and +/- 1 standard deviation denoted by the horizontal, black,
dashed lines. September of each year is represented by the vertical black, dashed

568                    lines.






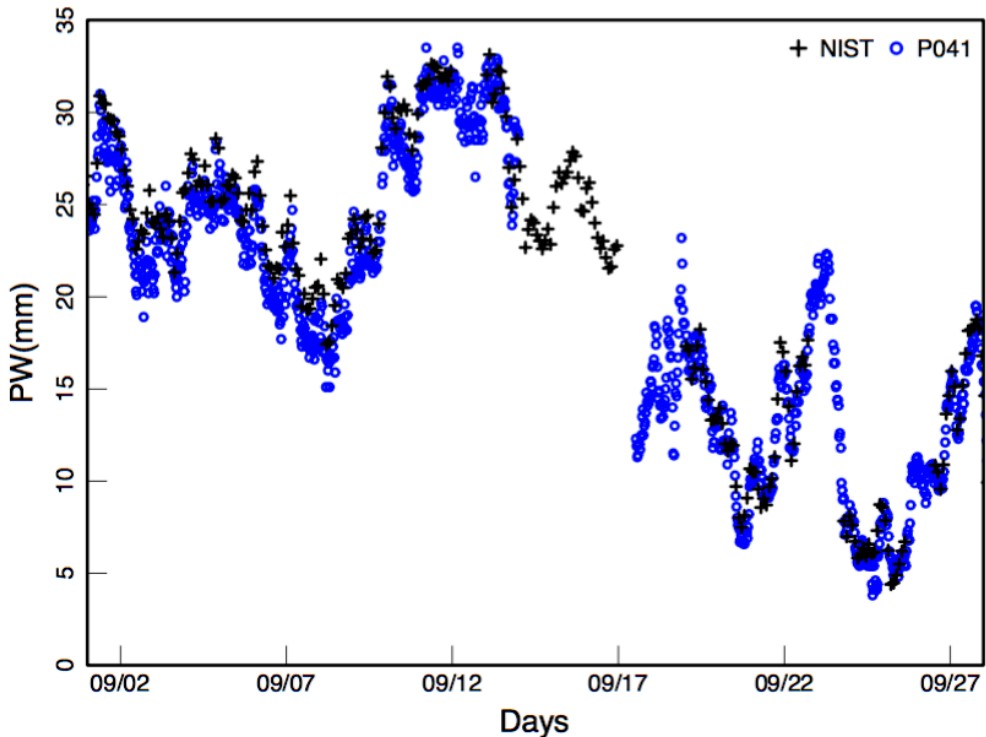


**Figure 3.** A time series of 30-minute GPS PW (station P041) from 1 – 28 September

2013. All times are in UTC.





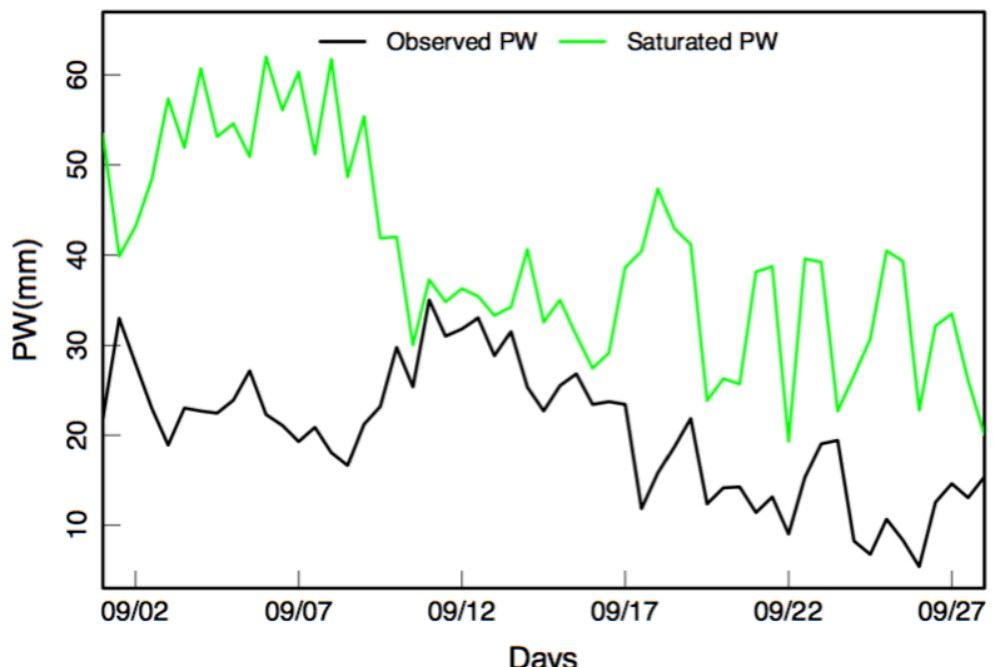


**Figure 4.** A time series comparison of observed radiosonde PW (black line) and saturated

PW (green line) for 1-28 September 2013 over Denver, Colorado. All

times are in UTC.






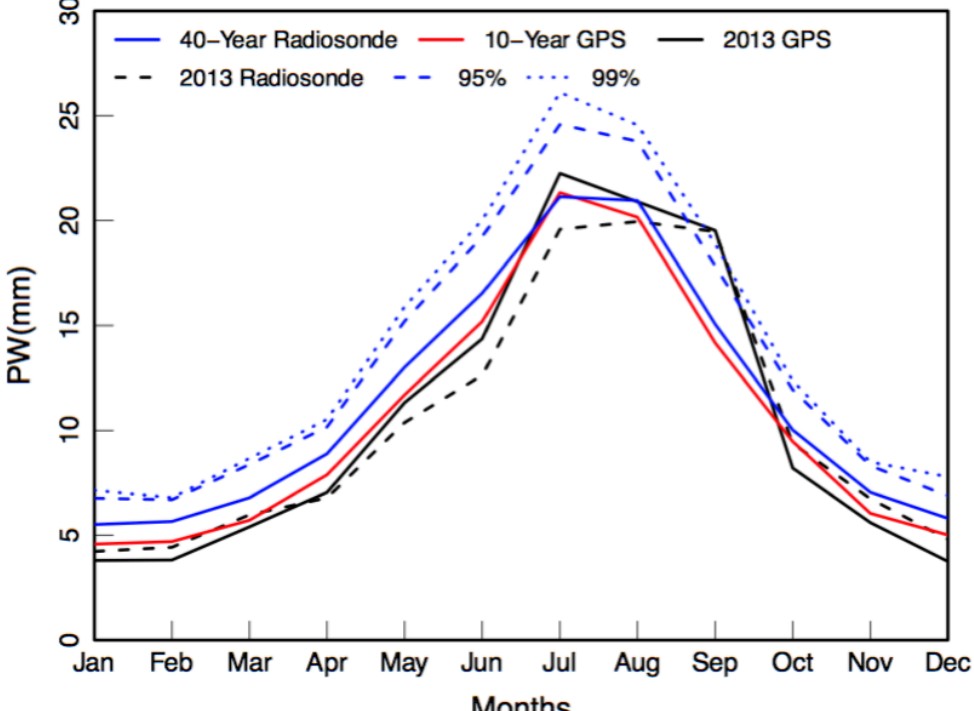


**Figure 5.** Monthly-averaged GPS PW (solid black line) and Radiosonde data (dashed

black line) for 2013 with the 10-year merged GPS PW dataset (solid red line) and the 40-

year averaged Radiosonde PW dataset (solid blue line). Additionally, there are the 95[th]

(dashed red line) and 99[th] (dotted red line) percentiles for 10 years of GPS data and the

95[th] (dashed blue line) and 99[th] (dotted blue line) percentiles for 40 years of Radiosonde

data.

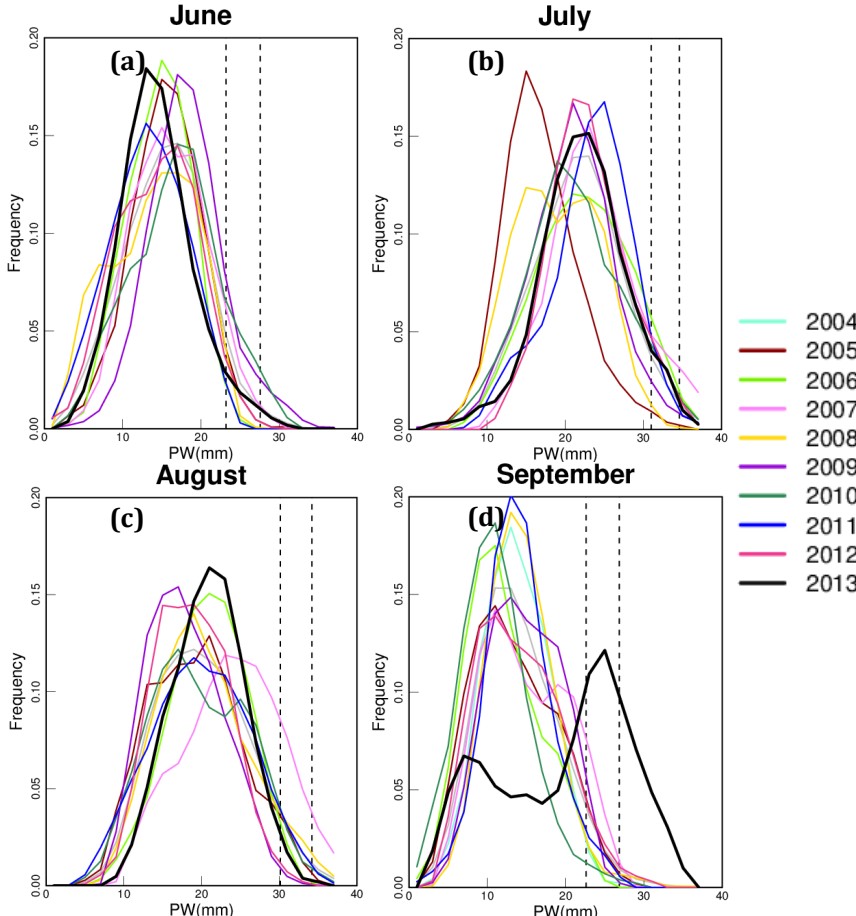

**Figure 6.** Statistical frequency distributions of GPS PW for June- September of 2004-
2013 with the 95th percentile for 10 years of each month of data denoted by the left-most
dashed line and the 99th percentile for 10 years of each month of data denoted by the
right-most dashed line.





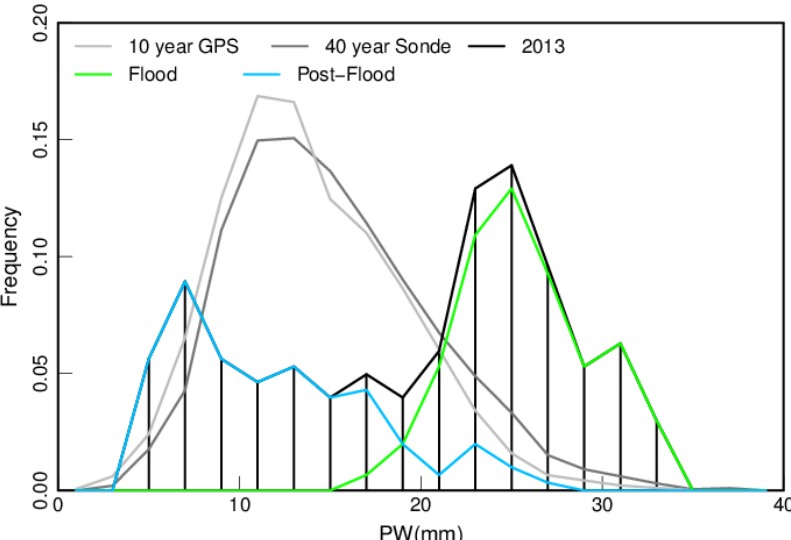

**Figure 7.** Statistical frequency distributions for the month of September with 2013 GPS

PW data over Boulder (black line), 40 years of climatologically-averaged radiosonde PW

data over Denver (dark grey line), and 10 years of climatologically-averaged GPS PW

data over Boulder (light grey line). September of 2013 GPS PW data was split into two

halves: 1-15 September 2013 (Flood; green line), and 16-30 September 2013 (Post-Flood;

blue line).






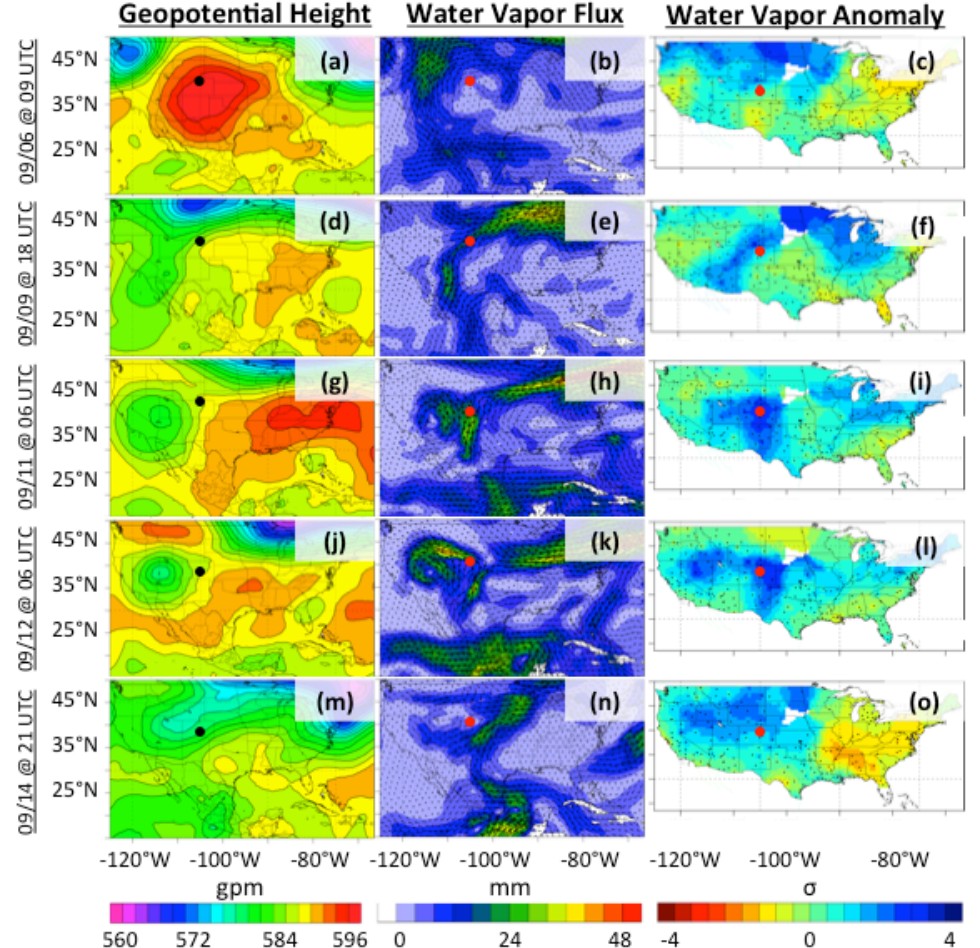

**Figure 8.** A comparison of NARR 3-hourly averaged 500 hPa geopotential height

(left column), NARR 3-hourly averaged integrated water vapor flux (center

column), and SuomiNet gridded standardized PW anomalies. Each row represents

a different time surrounding the 2013 Event.




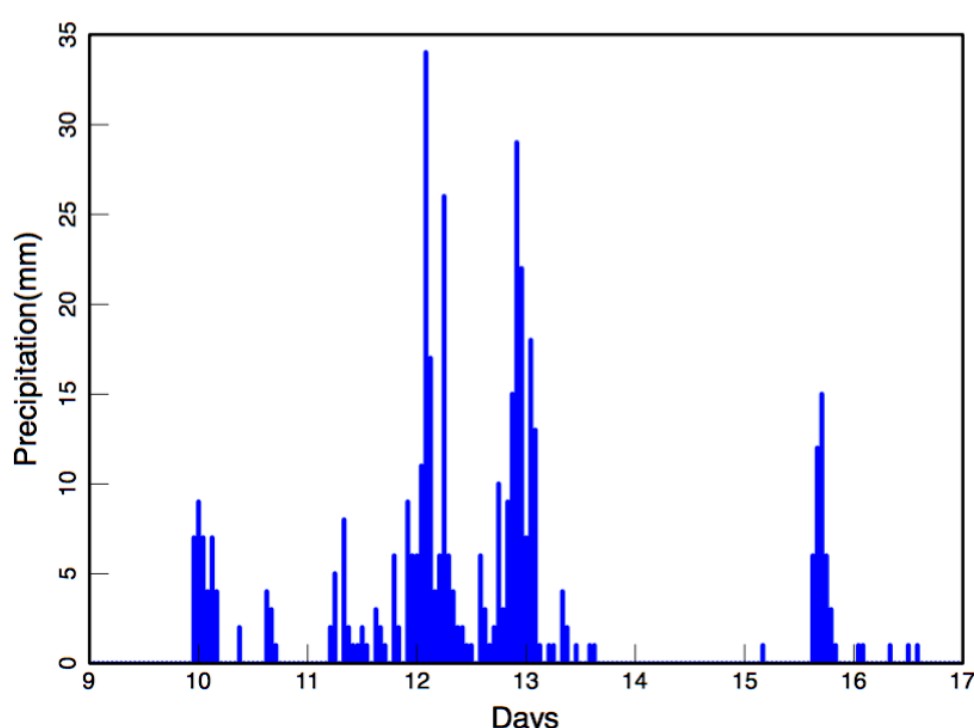

**Figure 9.** A time series of observed precipitation data from the rain gauge

UDFCD_4840 for 9-17 September 2013. All times are in UTC.

616

617