# Peer review of "Precipitable Water Characteristics during the 2013 Colorado Flood using Ground-Based"

_Atmospheric Measurement Techniques, 2016_

## Referee Comment (RC1) · D. Adams (Referee) · 12 Mar 2017

The revised manuscript is very close to acceptable for publication. Most of my concerns/criticisms of the submitted manuscript have been addressed.

I would just ask the authors to again address the question of saturation in the atmosphere. As I noted in my first revision, the atmosphere is simply not saturated from the surface to to the "top" of the atmosphere. You have to emphasize that you are making this assumption (which is TOTALLY unrealistic). It may be a question of semantics, but the atmosphere cannot saturate from the surface upwards, it would be radically unstable, and would not last a minute in that configuration. I ask you to at least note this assumption directly in the text.
comment

Interactive

AMTD

You don't have to change any of your calculations or figures, but just state explicitly that this is not a realistic assumption.

David K. Adams

---

## Referee Comment (RC2) · Anonymous Referee #3 · 15 Jun 2017

**General comments for authors**

In the manuscript titled "Precipitable Water Characteristics during the 2013 Colorado Flood using Ground-Based GPS Measurements" the authors present some studies about the PW variation during the most severe event of precipitation observed in Colorado State, using the GPS, reanalysis and radiosonde data. The main objective is to examine the characteristics of atmospheric PW during the 2013 Colorado Flood, data with a high spatial and temporal resolution was needed to resolve features within the event. However in my evaluation the present version of the manuscript does not demonstrate the potential benefits offered by GPS data to identify and PWV characteristics associated with severe event of precipitation. Two scientific questions are presented: (1) What were the characteristics of PW surrounding this event? and (2) Where did the moisture for the 2013 event originate? In (1) was not done a analysis of the GPS-PW before, during and after the precipitation event as promised. The analysis of climatological data is centered in a only figure, which the presentation is terrible to see the results. In (2) the transport of humidity is evaluated using the reanalysis products and water vapor anomaly from GPS data. The GPS-PW is a secondary information and some times it is erroneously used to evaluate the humidity flux. The high temporal resolution from PW GPS are mentioned but is not explored to characterize the PW variation during the severe precipitation event. I am not able to see in this manuscript how GPS data provide additional information than the radiosondes and reanalysis products. The most important results were obtained to use the radiosondes (figure 4 and 5) and reanalysis data (figure 8). Besides, this manuscript:

- (a) not presented the used methodology to obtain PW estimates from GPS data, as required by readers of this Special Issue;

- (b) the bibliographic revision is poor, which not mention the previous study about the GPS applied to PW monitoring and severe precipitation;

- (c) The precipitation time series is not presented by suitable way;

- (d) There are many technical corrections and imprecision in the text and presentation of results of this manuscript, which demonstrated lack of attention in the elaboration.

More details about each one these points are presented below in the section "Specific comments for authors". Summarizing I think this manuscript can not be published in the Special Issue: Advanced Global Navigation Satellite Systems tropospheric products for monitoring severe weather events and climate from AMT.

**Specific comments for authors**

**(I) General aspects of the manuscript:**

1) The motivation of the paper are not suitable and should be improved. I do not agree that there are not previous work that examine the characteristics of the PW and precipitation, although the number of this works is not expressive, some work should be mentioned and the results most important should be discussed in the introduction. No previous work was discussed. By the way, the number of the works referred in the text is very low. There are several research groups in different parts of the globe involved in this theme.

2) The climate change and relationship with severe precipitation are presented in introduction section out of context and a speculative discourse. This could be discussed in final section referring the other papers for mention the of possible link between these themes.

3) The authors state (lines 160-162): "... better understand the contributions of PW to an extreme precipitation event with the objective to someday apply these results to future research incorporating a wider variety of events." This statement can be the final aim of this research about GPS-PW. However, they are not clear how the GPS data can help in this aim, based on the results presented in this paper. GPS data are not suitable explored.

4) The aim of the work is examine the characteristics of PW during the precipitation event of 2013 and an analysis of the climatological series of PW is used. This analysis,

presented in Fig. 2, is poor and the quality (the size or zoom selected) not make evident the abnormality of this event. In my opinion this climatological study (as organized in this figure) does not help the examine the characteristics of the event. Besides, the PW time series shows that there are many similar values to observed in September of 2013 and severe precipitation was not observed (for example see September of 2011).

5) In this Special Issue, all details about the methods and options selected in the GPS data processing to obtain Zenithal Tropospheric Delay should be presented and discussed, as well as about the conversion in PW estimates. The current version of the manuscript discusses briefly the used methodology to obtain PW and only an paper is referred (Ware et al. 2000). Other more recent works should be used because the methodology employed in the GPS data processing have been improved in the last years. There are many aspects and options that would be taken into consideration to obtain PW, which significantly impact in the quality and behavior of PW time series from GPS data. For example: what kind of products for orbit and clock for satellite was used and what is the sampling of these products? Elevation-Dependent Weighting used for GPS observations was used? Was tropospheric models used for ZWD time evolution constraint as a random walk process? Were tropospheric gradients estimated? If yes what are the constraints of temporal evolution of these two parameters? Several specific works should be referred in this description.

6) The list of GPS stations used in this paper is not suitably presented, which are used in several parts of manuscript (i. e. figures and text) but the GPS stations are not described with more important information presented. For example, the Figure 1 shows the geographic localization of the GPS stations, but they are not mentioned in the text. Line 211: "This region encompasses six SuomiNet stations and two IGS stations (Fig. 1a)". Which stations were used? The figure 1a not present any station. A table with

more relevant information about the list of used GPS stations in this study should be presented in the beginning of the section 2.

**(II) Analysis of data and interpretation of results**

7) The precipitation time series should be better explored in this study. Although this information is crucial to characterize the GPS-PW variation before, during and after the precipitation severe event, the precipitation values are presented separately from PW values in a last figure of the manuscript (Figure 9). These data are critical for this study and should be better discussed in term of intensity and relationship with PW oscillations. Separating the time series of precipitation and GPS-PW the authors committed a serious mistake, which penalize the analysis of results and hinders to reach the proposed aims. The precipitation time series should be presented in Figs 3, 4 and 5, at least.

8)The Figure 2 shows PW values larger than the observed values in 2013 September (e. g. September of 2011) and it was not observed intense precipitation. This fact should be discussed.

9) Lines 283-294: The description of the results presented by Fig. 3 is very poor, which describe the period where the PW decreases and increase. The precipitation time series in this analysis should be interesting and it would help the analysis of those results.

10) Figure 4, I agree with David Adams that "fully saturated atmosphere, i.e. 100 relative humidity from the surface up to 300 hPa." can not be accepted and this analysis should be completely redone. I don't understand why the GPS-PW values in high temporal resolution are not explored in this analysis, which could be very much rich.

11) In the analysis of results in Section 3.2 the authors try to demonstrate the abnormality of PW values during September of 2013, when precipitation severe event was observed. I can not understand the reasons why in this analysis monthly-averaged values are used (Figure 5). The reported values of the monthly-averaged for September 2013 is the largest, which is the expected result above of normal. However, this analysis using monthly-averaged is not able to characterize the PW oscillation before, during and after the severe precipitation, as is proposed in the introduction section.

12) In section 3, GPS-PW in high temporal resolution are not explored before, during and after the severe event of precipitation, consequently the authors did not demonstrate the additional benefits obtained with GPS data than the usage of the other techniques of water vapor measurement. The same study can be carried out using the radiosonde data, which not justify the publication of this manuscript in this Spacial Edition about GNSS-PW estimates

13) In the analysis of water vapor transport, a bibliographic revision of previous work is done and results are reported. A similar analysis is done in the current version of the manuscript, using reanalysis data and water vapor anomaly from SuomiNet. The reported results not make evidence about the contribution of GPS data to corroborate with the results reported by these previous works.

14) In the analysis of water vapor transport (Figure 8) the selection of the time steps are aleatory or opportune without objective justification for the definition of these time steps. Why are these time steps used in different hours of the day? A figure with GPS-PW and precipitation (unacceptable lack in this study) would be used to justify this choice.

15) In the analysis of results the anomaly fields of water vapor from SuomiNet PW data (Figure 8) are used to indicate more drought or wetter atmospheric condition. It is a mistake because negative anomaly can not indicate a drier condition, but lower values than the climatological average. This is done in the line 391 and other parts of this manuscript

**(III)Technical corrections and imprecision**

16) Line 211: The Fig1.a is mentioned when the correct should be Fig.1b, which it is the correct plot that shows the GPS stations.

17) Lines 294-297: This comment about the humidity transport should be in the final of the subsection 3.2, before section 4.
18) Line 313: 2013 PW monthly averages were consistently lower than climatology until June and not July. In July the PW monthly averages were larger than climatology. .

19) Line 560: The NISU station is mentioned in the caption of the Figure 1, but I can not see this station in the plot of this figure. This station is used in others parts of the manuscript but: which are the information of this station? See item 6 above.

20) Line 490: The number of previous work referred in this manuscript is very low. There are many important papers about this theme that should be included in this study. No paper from AMT and EGU were mentioned here.

21) Line 550: The rain values presented in the Table 1 are in inches, and should be converted by mm, because the PW is presented in mm and the comparison is more direct when same unit are used.

22) Line 556: The Fig. 1a shows the preliminary precipitation accumulation values for Colorado, but this field is not mentioned in the manuscript. The plot 1.b of this figure is terrible to see the geographic localization of the GPS stations and coordinates are not expressed in this map.

23) Line 563: The Sept not should be in GPS station legend of the Figure 2.

24) Line 563: The figure 2 is not suitable for analysis of the results about the PW oscillation during the 2013 September and to compare with other Septembers. This figure should be significantly improved. The results are confused and they turn an

arduous task to affirm some conclusion.

25) Lines 573-587: The precipitation data should be in Figures 3, 4 and 5.

26) Line 574: The term "all times are in UTC" are not correct because the time in the figure is in Days. The same for the figures 3 4 and 9.

27) Line 586: In the figure 5 it is mentioned the 95th (dashed red line) and 99th (dotted red line) percentiles for 10 years and these lines are not showed.
* * *

---

## Author Comment (AC1) · 13 Jul 2017

**1. "I would just ask the authors to again address the question of saturation in the atmosphere. As I noted in my first revision, the atmosphere is simply not saturated from the surface to the "top" of the atmosphere. You have to emphasize that you are making this assumption (which is TOTALLY unrealistic). It may be a question of semantics, but the atmosphere cannot saturate from the surface upwards, it would be radically unstable, and would not last a minute in that configuration. I ask that you at least note this assumption directly in the text. You don't have to change any of your calculations or figures, but just state explicitly that this is not a realistic assumption."**

We thank the reviewer for reminding us of this point. The sentence "Note that a fully saturated atmosphere is an unrealistic assumption for a real atmosphere, but can be used for a simplified comparison" was added to Lines 306-308 to clarify that our assumption was unrealistic.

---

## Author Comment (AC2) · 13 Jul 2017

**General aspects of the manuscript:**

1. **The motivation of the paper are not suitable and should be improved. I do not agree that there are not previous work that examine the characteristics of the PW and precipitation, although the number of this works is not expressive, some work should be mentioned and the results most important should be discussed in the introduction. No previous work was discussed. By the way, the number of the works referred in the text is very low. There are several research groups in different parts of the globe involved in this theme.**
   Lines 147-151 were changed to be more specific and add in the only source with similar research who we have any knowledge of. We would appreciate if the reviewer would provide examples of prior work and also be more specific on why they think "the motivation is not suitable."

2. **The climate change and relationship with severe precipitation are presented in introduction section out of context and a speculative discourse. This could be discussed in final section referring other papers for mention the of possible link between these themes.**
   We do not agree with the reviewer's comment that this was "out of context and a speculative discourse." If extreme precipitation events such as the one studied here are rare, it might not be that important to study them. The point we were making by including this relationship is that as the climate is warming, we will see more of these events. Therefore, it is essential to study past events, understand physical processes, and improve the models to thus improve the prediction of such events in the future. As a result of improved understanding and prediction, more lives and properties can be saved.

3. **The authors state (lines 160-162): "... better understand the contributions of PW to an extreme precipitation event with the objective to someday apply these results to future research incorporating a wider variety of events." This statement can be the final aim of this research about GPS-PW. However, they are not clear how the GPS data can help in this aim, based on the results presented in this paper. GPS data are not suitable explored.**
   This sentence was revised to make our points clearer. It is important to study one particular event that had high impacts, but it is also essential to understand whether the results are applicable to other events which occurred at other times or in other regions. With regard to how GPS data can help in this aim, see our replies below on the importance of the high temporal resolution of GPS PW data. Also, we do not understand what the reviewer meant by the comment, "GPS data are not suitable explored."

4. **The aim of the work is examine the characteristics of PW during the precipitation event of 2013 and an analysis of the climatological series of PW is used. This analysis, presented in Fig. 2, is poor and the quality (the size or zoom selected) not make evident the abnormality of this event. In my opinion this climatological study (as organized in this figure) does not help the examine the characteristics of the event. Besides, the PW time series shows that there are many similar values to observed in September of 2013 and severe precipitation was not observed (for example, September of 2011).**

The main purpose of this figure was to show how we combined the data from five GPS sites to create a 10-year climatology and show no obvious discontinuity between stations at the adjoining boundaries. Figure 2 also serves as a preliminary comparison of PW in September of 2013 to previous Septembers. We have revised the plot to make it taller and show the September 2013 abnormal values more clearly. This figure also shows the seasonal variation observed in the Front Range region of Colorado. If the reviewer would once again examine this figure, they would see that high PW values extend into September only in 2013 (as was stated in Lines 279-280), and not in 2011.

5. **In this Special Issue, all details about the methods and options selected in the GPS data processing to obtain Zenithal Tropospheric Delay should be presented and discussed, as well as about the conversion in PW estimates. The current version of the manuscript discusses briefly the used methodology to obtain PW and only an paper is referred (Ware et al. 2000). Other more recent works should be used because the methodology employed in GPS data processing have been improved in the last years. There are many aspects and options that would be taken into consideration to obtain PW, which significantly impact in the quality and behavior of PW time series from GPS data. For example: what kind of products for orbit and clock for satellite was used and what is the sampling of these products? Elevation-Dependent Weighting used for GPS observations was used? Was tropospheric models used for ZWD time evolution constraint as a random walk process? Were tropospheric gradients estimated? If yes what are the constraints of temporal evolution of these two parameters? Several specific works should be referred to in this description.**

As PW processing techniques were not the focus of this particular study, an in-depth description would have brought the paper off-topic. Also, the processing details for each dataset have been thoroughly described in previous research. For SuomiNet data, which is processed by the COSMIC group at UCAR (and not the authors), the paper describing their processing technique is Ware et al. (2000) (Lines 192-196). The first two authors were, hoever, instrumental in processing the two-hourly, long-term PW dataset from IGS data (Lines 183-191). For this dataset, there are various papers cited within this manuscript which describe the processing techniques used in great detail (Lines 183-184). Any further description of the processing techniques would have been beyond the focus of this study.

6. **The list of GPS stations used in this paper is not suitably presented, which are used in several parts of manuscript (i. e. figures and text) but the GPS stations are not described with more important information presented. For example, the Figure 1 shows the geographic localization of the GPS stations, but they are not mentioned in the text. Line 211: "This region encompasses six SuomiNet stations and two IGS stations (Fig. 1a)". Which stations were used? The figure 1a not present any station. A table with more relevant information about the list of used GPS stations in this study should be presented in the beginning of the section 2.**
Line 211 was changed to read "Fig. 1b" instead of "Fig. 1a." A list of station data was also added as Table 2.

**Analysis of data and interpretation of results:**

7. **The precipitation time series should be better explored in this study. Although this information is crucial to characterize the GPS-PW variation before, during and after the precipitation severe event, the precipitation values are presented separately from PW values in a last figure of the manuscript (Figure 9). These data are critical for this study and should be better discussed in term of intensity and relationship with PW oscillations. Separating the time series of precipitation and GPS-PW the authors committed a serious mistake, which penalize the analysis of results and hinders to reach the proposed aims. The precipitation time series should be presented in Figs 3, 4 and 5, at least.**
As per the reviewer's request, Figures 3 and 9 were combined into the new Figure 3. A brief outline of the precipitation timeline during the flood was added.

8. **The Figure 2 shows PW values larger than the observed values in 2013 September (e. g. September of 2011) and it was not observed intense precipitation. This fact should be discussed.**
The seasonal oscillation observed in Figure 2 is discussed in Lines 273-276. The peaks represent Colorado's wet season. As was stated in response to Comment #4, the extension of moisture observed in September of 2013 is not observed in September of 2011. The peak you might be mistaking for being in September of 2011 was actually at the end of August. Fig. 2 has been improved in the revised manuscript and the new figure shows the abnormal values in September of 2013 more clearly.

9. **Lines 283-294: The description of the results presented by Fig. 3 is very poor, which describe the period where the PW decreases and increase. The precipitation time series in this analysis should be interesting and it would help the analysis of those results.**
Precipitation data was added to Figure 3 and description of this was added to the manuscript, as was stated in response to Comment #7.

10. **Figure 4, I agree with David Adams that "fully saturated atmosphere, i.e. 100 relative humidity from the surface up to 300hPa." can not be accepted and this analysis should be completely redone. I don't understand why the GPS-PW values in high temporal resolution are not explored in this analysis, which could be very much rich.**
David Adams suggested that we alter the description of the graph slightly to state that our assumption was unrealistic. Redoing this entire section is not an option, as we have already finished two rounds of reviewer comments. Had the reviewer responded with this suggestion in the first round, there may have been time to devise a new research method for this particular graph.

11. **In the analysis of results in Section 3.2 the authors try to demonstrate the abnormality of PW values during September of 2013, when precipitation severe event was observed. I can not understand the reasons why in this analysis monthly-averaged values are used (Figure 5). The reported values of the monthly-averaged for September 2013 is the largest, which is the expected result above of normal. However, this analysis using monthly-averaged is not able to characterize the PW oscillation before, during and after the severe precipitation, as is proposed in the introduction section.**
Monthly-averaged values were used because the usage of every point would have resulted in a noisy, indecipherable graph. The use of monthly-averaged values resulted in a more easily-deciphered graph that showed us just how abnormal the PW values during the flood were that they pushed the monthly average above the 99$^{th}$ percentile.

12. **In section 3, GPS-PW in high temporal resolution are not explored before, during and after the severe event of precipitation, consequently the authors did not demonstrate the additional benefits obtained with GPS data than the usage of the other techniques of water vapor measurement. The same study can be carried out using the radiosonde data, which not justify the publication of this manuscript in this Special Edition about GNSS-PW estimates.**
The high resolution GPS-PW characteristics around the timing of the flooding event were discussed on Lines 283-297. Figures 3 and 8 showed the advantages of using GPS PW data for the detection of abnormal atmospheric moisture amounts which could lead to heavy precipitation events.

13. **In the analysis of water vapor transport, a bibliographic revision of previous work is done and results are reported. A similar analysis is done in the current version of the manuscript, using reanalysis data and water vapor anomaly from SuomiNet. The reported results not make evidence about the contribution of GPS data to corroborate with the results reported by these previous works.**
We do not fully understand the reviewer's points here. Standardized anomalies of GPS-PW data reflected the patterns of moisture flux shown with the reanalysis data. This can serve as the basis to overlay 750 hPa winds on the PW anomalies to determine moisture transport in future studies.

14. **In the analysis of water vapor transport (Figure 8) the selection of the time steps are aleatory or opportune without objective justification for the definition of these time steps. Why are these time steps used in different hours of the day? A figure with GPS-PW and precipitation (unacceptable lack in this study) would be used to justify this choice.**
   As was stated in Lines 384-386, the times were chosen based on their proximity to rapid fluctuations in PW. The precipitation data from the previous Fig. 9 have been added to the new Fig. 3.

15. **In the analysis of results the anomaly fields of water vapor from SuomiNet data (Figure 8) are used to indicate more drought or wetter atmospheric condition. It is a mistake because negative anomaly can not indicate a drier condition, but lower values than the climatological average. This is done in the line 391 and other parts of this manuscript.**
   When the authors refer to "drier conditions," they are comparing the anomalies observed in the panels to one another. Therefore, when the anomaly is lower in one panel compared to the previous one, it can be assumed that the atmosphere is drier in that panel than the previous panel.

**Technical corrections and imprecision:**

16. **Line 211: The Fig1.a is mentioned when the correct should be Fig. 1b, which it is the correct plot that shows the GPS stations.**
   As was mentioned in reply to Comment #6, this was altered.

17. **Lines 294-297: This comment about the humidity transport should be in the final subsection 3.2, before section 4.**
   We do not agree with the reviewer. Section 3.1 is about the temporal variability of PW.

18. **Line 313: 2013 PW monthly averages were consistently lower than climatology until June and not July. In July the PW monthly averages were larger than climatology.**
   This is a question of semantics and misinterpretation. The phrasing "up until July" means "prior to July."

19. **Line 560: The NISU station is mentioned in the caption of the Figure 1, but I can not see this station in the plot of this figure. This station is used in others parts of the manuscript but: which are the information of this station? See item 6 above.**
   This station is collocated with the station NIST and the label was mistakenly left out. This has been fixed in the latest version of the manuscript.

20. **Line 490: The number of previous work referred in this manuscript is very low. There are many important papers about this theme that should be included in this study. No paper from AMT and EGU were mentioned here.**
   We would be most appreciative if the reviewer could suggest some previous works for us to read through. We have included what was found through numerous literature searches and we did not intentionally leave out papers from AMT or EGU.

21. **Line 550: The rain values presented in Table 1 are in inches, and should be converted to mm, because the PW is presented in mm and the comparison is more direct when the same unit are used.**
   The rainfall total units in Table 1 were changed to mm.

22. **Line 556: The Figure 1a shows the preliminary precipitation accumulation values for Colorado, but this field is not mentioned in the manuscript. The plot 1.b of this figure is terrible to see the geographic localization of the GPS stations and coordinates are not expressed in this map.**
   This was added as Table 2.

23. **Line 563: The Sept should not be in GPS station legend of the Figure 2.**
   This was removed per the reviewer's request.

24. **Line 563: The figure 2 is not suitable for analysis of the results about the PW oscillation during the 2013 September and to compare with other Septembers. This figure should be significantly improved. The results are confused and they turn an arduous task to affirm some conclusion.**
   Fig. 2 is improved. We have also clarified the purpose of this figure in the responses above to help the reviewer understand it better.

25. **Lines 573-587: The precipitation data should be in Figures 3, 4 and 5.**
   Fig. 3 and 9 have been combined. This was addressed in Comments #7 and 9.

26. **Line 574: The term "all times are in UTC" are not correct because the time in the figures is in Days. The same for the figures 3 4 and 9.**
   These were removed per the reviewer's request.

27. **Line 586: In the figure 5 it is mentioned the 95[th] (dashed red line) and 99[th] (dotted red line) percentiles for 10 years and these lines are not showed.**
   This line was removed from the figure description as it had been accidentally left in from previous edits.